# Collagen-Based Bioactive Bromelain Hydrolysate from Salt-Cured Cod Skin

Ezequiel R. Coscueta *[ID], María Emilia Brassesco [ID] and Manuela Pintado *

CBQF—Centro de Biotecnologia e Química Fina—Laboratório Associado, Universidade Católica Portuguesa, Escola Superior de Biotecnologia, Rua Diogo Botelho 1327, 4169-005 Porto, Portugal; mbrassesco@ucp.pt
* Correspondence: ecoscueta@ucp.pt (E.R.C.); mpintado@ucp.pt (M.P.); Tel.: +351-225-580-001 (ext. 8047) (E.R.C.)

**Abstract:** Considerable amounts of fish processing by-products are discarded each year. About 30% of this material may be skin and bone. Fish skin has more than 80% of its total protein content as collagen. Furthermore, in recent years, there has been a growing demand for collagen-based peptides due to their beneficial health effects. So, the objective of the present study was to optimise the obtaining bioactive hydrolysates from salt-cured cod skin using the protease Bromelain at 0.5% ($w/w$) concentration. This study developed a sustainable process that consumes less time and energy and uses an alternative source as raw material. In addition, bromelain allows hydrolysates with important antioxidant (ORAC, 514 μmol Trolox Equivalent/g protein) and antihypertensive activities (inhibition of ACE, $IC_{50}$ of 166 μg protein/mL) as well as excellent biocompatibility with dermal and subcutaneous cells.

**Keywords:** Atlantic cod skin; enzymatic hydrolysis; collagen; fishery by-products; bioactive peptides; bromelain; antioxidant; antihypertensive





## 1. Introduction

Collagen is a structural protein present in different animal tissues. Its technological properties have been fundamental to various industries, and currently, its interest has increased due to its potential bioactive properties. Partial hydrolysis of native collagen produces gelatine, which is the usual collagen prepared industrially. Collagen and gelatine have different industrial applications; among the most well-known are the food, cosmetic, and biomedical industries [1]. In recent years, the demand for collagen-based peptides has grown dramatically. This is mainly due to the beneficial effects of these compounds on the skin. Oral ingestion of collagen-based peptides (generally gelatine hydrolysates) has been seen to promote collagen synthesis in the skin, increasing the size of collagen fibrils in the dermis. This is believed to improve skin hydration and prevent wrinkles [2]. Collagen-derived peptides also have antihypertensive and antioxidant activities that are associated with low molecular weight peptide structures [3]. Collagen-rich hydrolysates can be produced by controlled thermal or enzymatic hydrolysis [4]. However, enzymatic hydrolysis is preferred because it produces the material in its native state and needs less time and milder conditions to obtain lower molecular weight peptides [5]. Several enzymes have been used for the hydrolysis process [6–9]. However, the protein hydrolysates prepared using bromelain showed higher antioxidant capacity, desired functional properties, and excellent industrial applications [10–12]. The industrial production of collagen-derived peptides requires two coupled steps [13]. In the first, collagen is extracted and purified. In the second step, the peptides are obtained by enzymatic hydrolysis, they are sterilised, and finally they are dried [13]. Therefore, collagen extraction is time and energy consuming [2].

Classically, collagen is obtained from skins of mammalian and filamentous tissues but with several negative aspects. In recent years, due to the outbreak of animal diseases such as Bovine Spongiform Encephalopathy, Transmissible Spongiform Encephalopathies, and Foot and Mouth Disease, the search for alternative sources of collagen has increased [14]. The

increase in fish consumption per capita and the consequent increase in fisheries by-products in the past two decades led academia and the industry to explore marine by-products as a reliable and economical source of many healthy and biologically active compounds such as collagen [15]. It includes marine fish, starfish, sponges, jellyfish, squid, etc. [16–19]. Fish residues may account for an average of 55% of the total fish weight; of this material, up to 30% may be skin and bone [17]. Fish skin has more than 80% of its total protein content as collagen. The work performed to date proves that marine fish may be a promising source.

This study aims to develop a new integrated methodology to produce a collagen-rich enzymatic hydrolysate, taking advantage of the salt-cured cod skin a by-product from cod industry abundant in countries such as Portugal.

## 2. Materials and Methods

### 2.1. Materials

Salt-cured cod skin (SCCS) from Atlantic cod (*Gadus morhua*) was kindly provided by Pascoal & Filhos S.A. The skin was obtained as a by-product before the cutting and processing fish meat to obtain different portions to sell or for ready to eat meals. The samples were transported at room temperature and stored at −20 °C before processing. Calfskin type-I collagen was purchased from Sigma-Aldrich (St. Louis, MO, USA), and molecular weight marker was purchased from NZYTech, Lisboa, Portugal. Cod skin collagen was purified by acid methodology in previous work [20]. For the enzymatic hydrolysis, the commercial enzymes collagenase from *Clostridium histolyticum* (CHC), which is a protease specific for collagen and the cysteine protease Bromelain from pineapple stem (BR) were purchased from Sigma-Aldrich (St. Louis, MO, USA) with a declared activity of ≥125 CDU/mg solid (CDU = collagen digestion units) and 3.98 units/mg protein (one unit release 1.0 μmol of *p*-nitrophenol from *n*-alpha-CBZ-L-Lysine *p*-nitrophenyl ester per minute at pH 4.6 at 25 °C), respectively. MEM non-essential amino acid solution was supplied by Sigma-Aldrich (Merck, Darmstadt, Germany). Dulbecco's Modified Eagle Medium (DMEM) high glucose and Penicillin-Streptomycin mixture were obtained from Lonza (Basel, Switzerland). Foetal bovine serum (FBS) was purchased from Biowest (Nuaillé, France). All the other reagents were of analytical grade and used without further purification.

### 2.2. Pre-Treatment of Skin

The pre-treatment of skin was carried out following the methodology of Arumugam et al. [21], with some modifications. First, the cod skins were washed with tap water 3 times at 25 °C to remove the salt, residual fat, and flesh of fish (by the action of high salt concentration) and then cut into small pieces. Then, those pieces were mixed with 0.1 M NaOH at 25 °C to remove non-collagenous proteins and pigments at a sample-to-solution ratio of 1:10 (*w/v*), about 4 h. The mixture was centrifuged (Hettich Universal 320R) at 3857 RCF for 15 min at 4 °C. The resulting solid residue was washed with distilled water 5 times until neutral (pH 7) at 25 °C and then centrifuged according to the conditions described above.

### 2.3. Preliminary Analysis

The obtaining of collagen-rich hydrolysate with a plant enzyme such as BR was evaluated compared with the activity of CHC under similar conditions (37 °C, pH 7.2) for 16 h.

### 2.4. Degree of Hydrolysis Analysis

The degree of hydrolysis (DH) was determined using the method described by Nielsen, Petersen and Dambmann (2001) [22], with some modifications [23].

### 2.5. Yield of Production

The yield was calculated as the percentage ratio of skin mass in gram to the hydrolysates mass obtained in g, all on a dry basis.

$$\text{Yield } (\%) = \frac{\text{Mass of the hydrolysate (g)}}{\text{Mass of the total skin hydrolysis (g)}} * 100 \tag{1}$$

For the mass of the hydrolysate, the mass equivalent to the saline content remaining in the solution, the residual acetate, was subtracted.

### 2.6. Protein Concentration

The protein concentration was measured by the Pierce BCA Protein Assay kit (Thermo Scientific, Waltham, MA, USA), using bovine serum albumin as standard.

### 2.7. Tricine-SDS-PAGE

Tricine-SDS-PAGE [24] was prepared using 6% stacking and 12% resolving gel. 1.0 mg/mL Calfskin type I collagen, initial time and 30 min of the hydrolysis with bromelain, collagenase, and control (without enzymes) were mixed in 0.5 M acetic acid, diluted $\frac{1}{2}$ with sample buffer, incubated at 80 °C for 20 min, and centrifuged at 21,382 RCF for 10 min. Twenty microliters of supernatants were loaded in gel. The electrophoresis ran at a constant voltage of 75 V for about 5 min followed by 150 V. After electrophoresis, the gel was stained in Coomassie blue G-250 (0.25%, *w/v*), staining solution for 1 h, and discoloured overnight. The final image was acquired with the ChemiDoc™ XRS+ and analysed by the Imaging System Image Lab™ Software Version 6.0.1.34.

### 2.8. Hydrolysis Optimisation

An integrative process to produce bioactive hydrolysates from SCCS was developed and optimised. The hydrolysis process was optimised following for each experiment the same steps described in Scheme 1 (Optimised process). A multifactor experimental arrangement was designed following the Box Behnken model. Three experimental factors were considered, and three responses were analysed. The factors evaluated were pH value ($X_A$), temperature ($X_B$), and hydrolysis time ($X_C$) and the selected response variables (Y) were the protein concentration, antioxidant activity, and antihypertensive activity. The design resulted in an arrangement of 15 treatments, which was executed in triplicate (45 runs) on successive days. The levels of the factors, coded as −1 (low), 0 (central point), and +1 (high), are shown in Table 1. The ratio of enzyme to the substrate was kept constant at 0.5% *w/w* solid.

**Table 1.** Levels of factors evaluated on the hydrolysis process optimisation.

| Factors | Low | High |
|:---:|:---:|:---:|
| $X_A$: pH | 4.0 | 7.2 |
| $X_B$: Temperature (°C) | 25.0 | 45.0 |
| $X_C$: Time (min) | 30.0 | 270.0 |

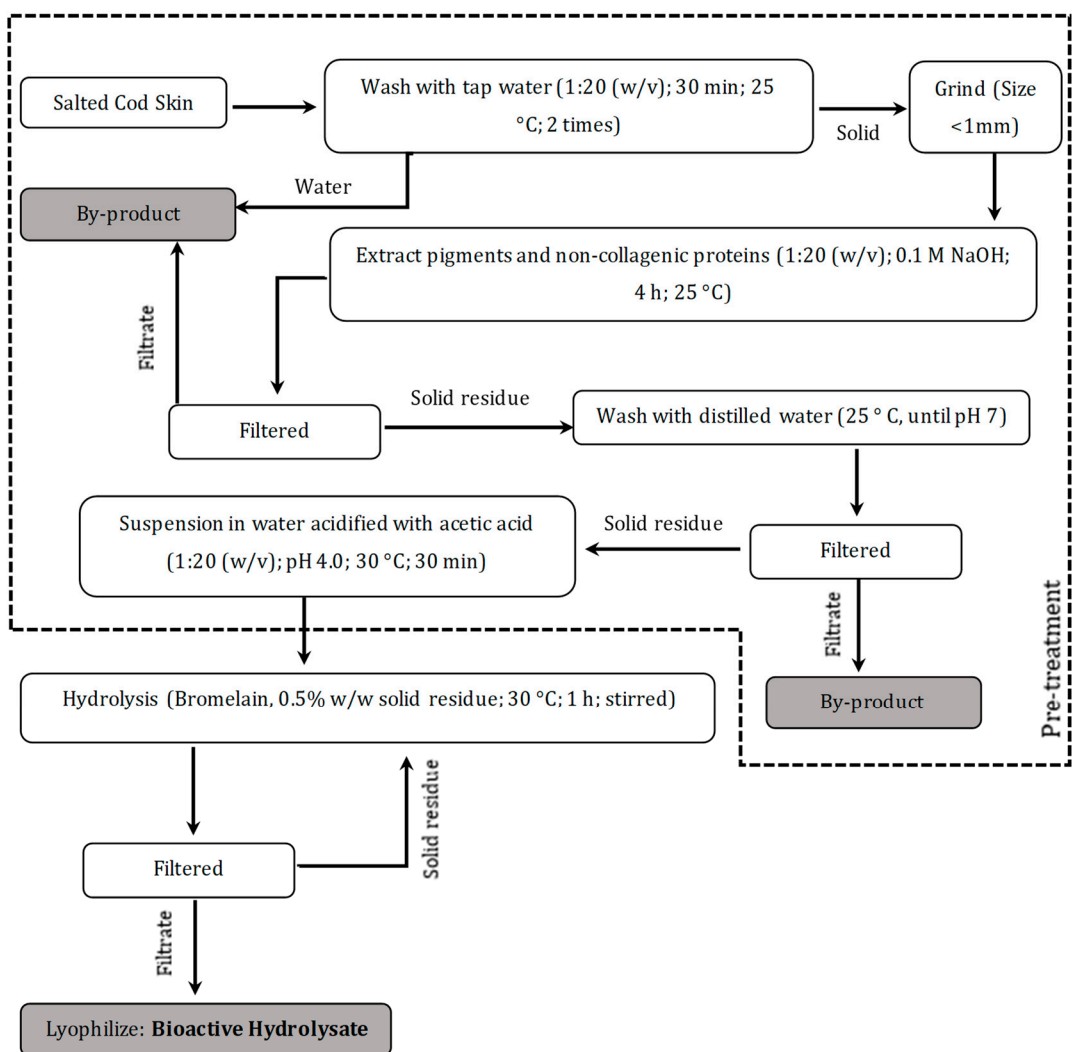

**Scheme 1.** Optimised process. Flow diagram of the optimal process proposed to produce the bioactive hydrolysates from SCCS.

### 2.9. Bioactivities

The analysis of the antioxidant activity was performed following the methodology described previously [25]. ORAC values were expressed as μmol TE (Trolox equivalent)/g protein. The antihypertensive activity was assayed by the inhibitory activity on the angiotensin-I converting enzyme (iACE) as described previously [26]. iACE was expressed as % inhibition for optimisation tests and as the concentration capable of inhibiting 50% of the enzymatic activity ($IC_{50}$) for the optimal hydrolysate. Both assays were realised with the multidetection plate reader Synergy H1 (BioTek Instruments, Winooski, VT, USA) controlled by the Gen5 BioTek software version 3.04.

### 2.10. In Vitro Biocompatibility

Two different cell lines were considered throughout this work, namely, mouse fibroblast cells—L929 (NCTC) (ECACC 85103115); and human keratinocyte—HaCaT (300493, CLS, Eppelheim, Germany). L929 and HaCaT cells were cultured in DMEM high glucose supplemented with 10% (*v/v*) FBS and 1% (*v/v*) penicillin-streptomycin. The tests were carried out following the same procedure that we previously reported [27]. The culture media were carefully removed and replaced with 1 mg hydrolysate $mL^{-1}$ (sterile filtered). The cytotoxicity was evaluated using the PrestoBlue™ HS Cell Viability assay (Thermo Scientific, Waltham, MA, USA), following the protocol described by the manufacturer.

Fluorescence was measured using a fluorescence excitation wavelength of 560 nm and an emission of 590 nm by the multidetection plate reader Synergy H1 (BioTek Instruments, Winooski, VT, USA) controlled by the Gen5 BioTek software version 3.04. The metabolic inhibition was determined as previously reported [27].

### 2.11. Statistical Analysis

2.11.1. Preliminary Analysis

Since in the data obtained, the response variables were recorded more than once on the same experimental unit, the analysis is considered as "repeated measures", applying then the procedure of Generalised Linear Models, GLM [28]. Each experiment was performed in duplicate.

The mean values were analysed statistically by one-way analysis of variance (ANOVA) followed by Tukey's post hoc test [29,30]. Separation of means was conducted by the least significant difference at the 5% probability level.

2.11.2. Optimisation

The process optimisation was carried out in triplicate of the design, expressing the results as mean values with standard deviations (SD). The responses (Y) were fitted to the following polynomial quadratic model:

$$Y = \beta_0 + \beta_A X_A + \beta_B X_B + \beta_C X_C + \beta_{A,A} X_A^2 + \beta_{A,B} X_A X_B + \beta_{A,C} X_A X_C + \beta_{B,B} X_B^2 + \beta_{B,C} X_B X_C + \beta_{C,C} X_C^2 + \varepsilon \qquad (2)$$

where $X_A$ and $X_B$ are the coded levels of the independent variables mentioned above; $\beta_0$, $\beta_i$, $\beta_{i,I}$, and $\beta_{i,j}$ are the regression coefficients for the independent term, the linear, quadratic, and binary interaction effects respectively; and $\varepsilon$, the residual error [31,32]. Following the polynomial model, surface and contour plots of each response were generated. The significance of the effects of each experimental factor was estimated for the model of each response. The regression models that best explained the variability of the data (best $R^2$) were derived from recalculating the initial models, just considering the significant effects. Finally, a multicriteria optimisation based on the Derringer desirability function [33] was applied to the results of the experimental design, expressing the desirability of each response value on a 0–1 scale.

The major statistical analysis was carried out with the aid of RStudio V 1.2.1335.

### 3. Results

#### 3.1. Preliminary Analysis

The DH determination is the key parameter that determines the functional and biological activity of resultant hydrolysates. In our case, the obtaining of collagen hydrolysate with a plant enzyme such as BR was evaluated in comparison with a known specific enzyme for collagen (CHC), being the used conditions the optimal for CHC [34] but not for BR. As a result, low molecular weight hydrolysates were obtained with a protein concentration that was not different between the two enzymes (1.5 mg/mL). As shown in Figure 1, the tendency of the DH was similar between both enzymes. However, under the conditions tested, BR did not reach CHC levels, which is justified by the conditions used for the experiments.

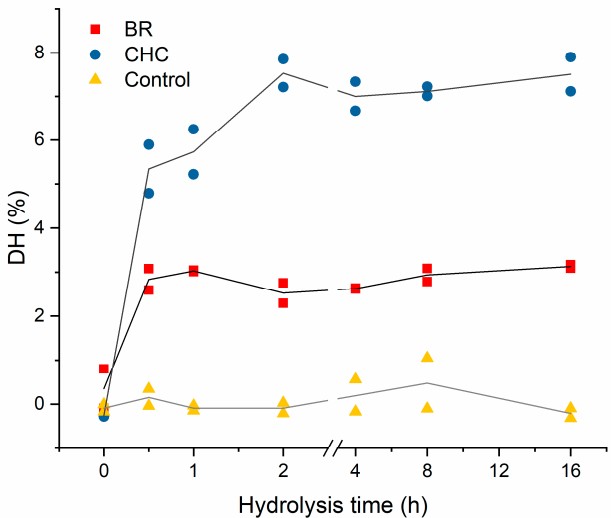

**Figure 1.** Degree of hydrolysis (DH). DH is expressed in percentage as a function of hydrolysis time expressed in hours. BR: bromelain; CHC: collagenase; Control: without enzyme.

Through the analysis of the electrophoresis (Figure 2), the bands at the initial time (with enzymes) and in the control (without enzyme addition) were compared to the Calfskin type I collagen, which allowed us to see that the main pattern corresponded to the typical for type I collagen. On the other hand, 30 min later of the hydrolysis processes, the principal bands corresponding to the collagen disappeared, whereas the control remained intact. These results confirmed the successful hydrolysis capacity of BR.

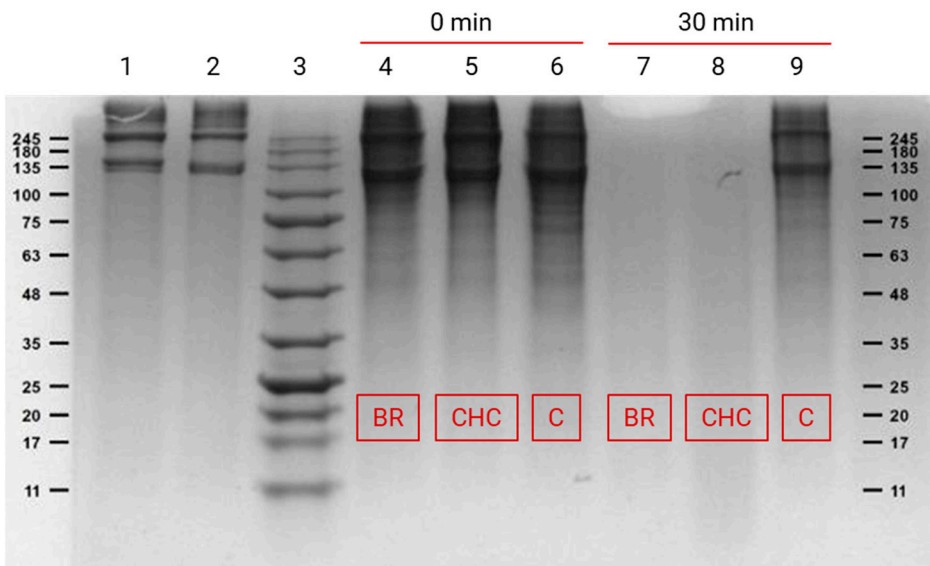

**Figure 2.** SDS-PAGE: Calfskin type I collagen (lane 1), collagen purified from cod skin by acid methodology (lane 2); molecular weight marker (lane 3); initial time of the hydrolysis with bromelain (lane 4), collagenase (lane 5) and control (lane 6); hydrolysis at 30 min with bromelain (lane 7), with collagenase (lane 8) and control (lane 9). Molecular weight in kDa.

### 3.2. Hydrolysis Optimisation

When we saw that BR was a good candidate to apply in the production of collagen hydrolysates, we proceeded to design a process that had to be optimised. For this, we carry out an experimental design as described in the methodology.

Supplementary Materials shows all the data acquired for the design and the complete multifactorial analysis for the 15 randomised treatments. The finally adjusted models explained the variability in a good way, as shown by their determination coefficients ($R^2$ in Figure 3). Figure 3 shows the standardised effects for the coefficients of the factors of the recalculated models (criterion explained in the methodology). We observed that temperature was a significant factor for the three responses considered. For protein extraction, increasing the temperature increased the response and an important maximising curvature effect.

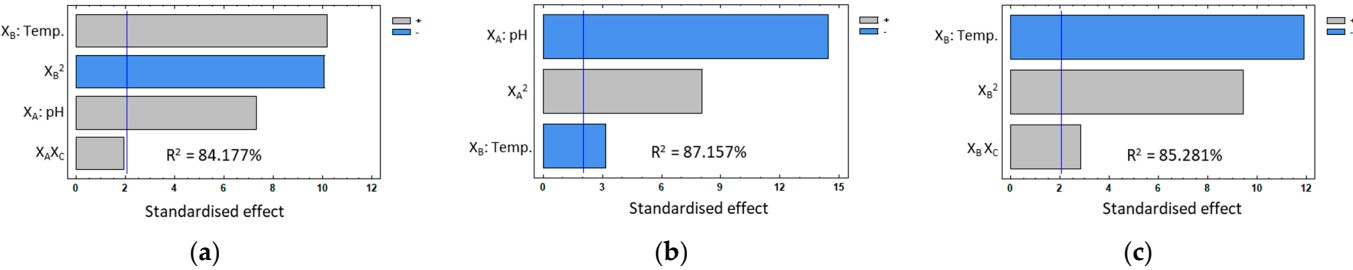

**Figure 3.** Analysis of the effects of the Box–Behnken factorial design. Pareto charts with standardised effects of three experimental factors, in decreasing order of importance (in absolute value) for the three responses: protein concentration (**a**); ORAC (**b**); and iACE (**c**). The vertical blue lines represent the threshold of significance ($p = 0.05$) for 32 degrees of freedom.

On the other hand, for bioactivities, the increase in temperature had a negative effect. The pH effect was significant both for the protein content and antioxidant activity but not for the antihypertensive activity. For protein extraction, increasing the pH increased the extractive capacity. On the contrary, in antioxidant activity, when the pH increased, the bioactivity decreased. In the case of the time factor, it was not significant for any of the responses analysed. However, its interaction with temperature was significant in the case of antihypertensive activity; its interaction with pH was considered in the case of the protein extraction (improved $R^2_{adj}$).

In a first step, each response was optimised, obtaining different conditions according to the response to maximise (Figure 4a–c; Table 2). Derringer's total desirability model was applied to carry out general optimisation. Thus, the condition that would allow for the achievement of a good joint maximisation of the responses was reached (Table 3). In the optimal global conditions, the antioxidant activity value (ORAC) is 514 µmol Trolox Equivalent/g protein) and the iACE value corresponds to an $IC_{50}$ of 59 µg protein/mL. From the optimal levels, the validation of the models was carried out, but with a slight modification, instead of 29.8 °C, 30.0 °C was applied, and the predictions were re-entered with these values. Table 3 shows the predicted values for these validation conditions and the observed values for an experiment done in a single execution (for this reason, it does not present any deviation). From the hypothesis test, we did not find significant differences for a confidence level of 5%. Therefore, the models are valid. Besides, by producing the hydrolysate in this condition, a yield of 44% was achieved. Furthermore, it should be mentioned that, for the hydrolysate produced for the validation, the iACE showed an $IC_{50}$ of 166 µg protein/mL. This value corresponds to a significant antihypertensive potential [23,26].

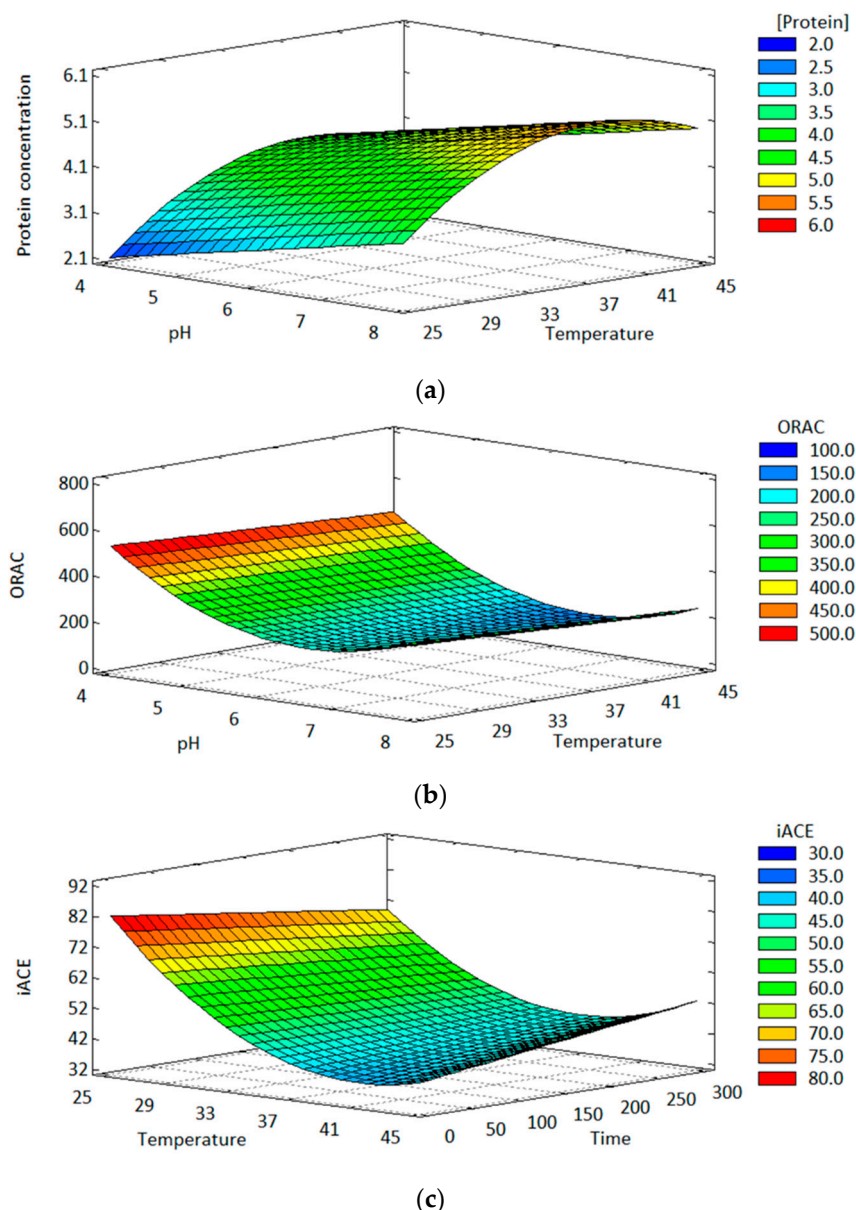

**Figure 4.** Response surface models. Each response is based on two experimental factors keeping the third factor at its central level. Charts for: protein concentration (**a**); ORAC (**b**); iACE (**c**).

**Table 2.** Factor's levels that optimise each response and the corresponding predicted optimal response value.

| Model | Factors † | | | Response ‡ |
|---|---|---|---|---|
| | $X_A$ | $X_B$ | $X_C$ | |
| Protein concentration | 7.2 | 38.5 | 270.0 | 5.343 |
| ORAC | 4.0 | 25.0 | 150.0 | 530 |
| iACE | 5.6 | 25.0 | 30.0 | 80.9 |

† Factors: pH ($X_A$); temperature ($X_B$) expressed in °C; time ($X_C$) expressed in min. ‡ Responses: protein concentration expressed in mg/mL; ORAC expressed in µmol TE/g protein; iACE expressed in % inhibition/(0.5 mg/mL protein).

**Table 3.** Derringer optimisation and model validation.

| Factor [†] | Optimal | |
|---|---|---|
| $X_A$ | 4.0 | |
| $X_B$ | 29.8 | |
| $X_C$ | 30.0 | |
| Optimal desirability: 0.516 | | |
| **Model validation** | | |
| Response [‡] | Predicted * | Observed |
| Protein concentration± | 3.3 ± 0.7 | 2.2 |
| ORAC | 517 ± 92 | 414 |
| iACE | 63 ± 11 | 59 |

[†] Factors: pH ($X_A$); temperature ($X_B$) expressed in °C; time ($X_C$) expressed in min. [‡] Responses: protein concentration expressed in mg/mL; ORAC expressed in µmol TE/g protein; iACE expressed in % inhibition/(0.15 mg/mL protein). * Values expressed as mean ± SD of three replicates.

### 3.3. Biocompatibility Assay

The biocompatibility of the hydrolysates was also tested at a protein concentration of 1 mg/mL in murine fibroblast cells (L929) and immortalised human keratinocytes (HaCaT). In the test, we found that the hydrolysate in the tested concentration was not cytotoxic for any of the cell lines. As shown in Figure 5, negative values prevail for both lines, indicating a possible stimulation of cellular metabolism.

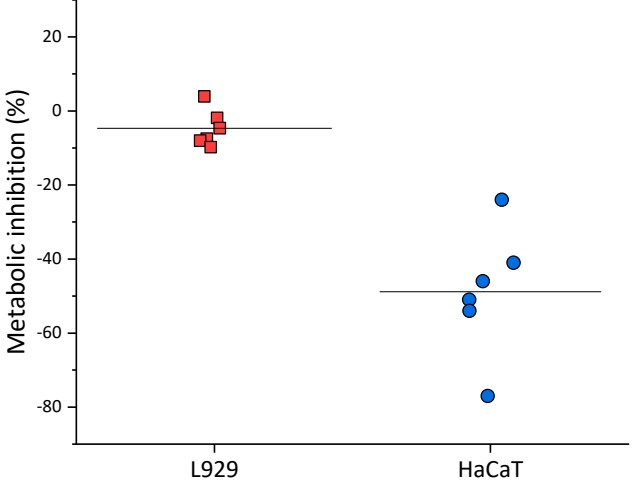

**Figure 5.** Metabolic inhibition of optimal hydrolysate with a concentration of 1 mg/mL for 24 h against two mammalian cell lines, namely: human keratinocyte HaCaT cells; and mouse fibroblast L929 cells. Each boxplot represents $n = 6$.

## 4. Discussion

This study aimed to develop a new integrated methodology to produce a collagen-rich enzymatic hydrolysate from SCCS sustainably and simply. On the one hand, enzymatic hydrolysis is the best way to hydrolyse fish skin without losing nutritional and healthy values [35]. On the other hand, the increased fish processing sector results in massive production of waste, which makes it a challenge to dispose of these materials to optimise value and reduce environmental impact [36]. So, in the first step, we compared the BR hydrolysis capacity to CHC, a specific protease for collagen. BR is a vegetal cysteine endopeptidase with broad specificity used for many industrial applications [12] and was already proved to be effective in hydrolysed collagens [37]. BR was also successfully applied to collagen extraction from the skin of bigeye tuna (*Thunnus obesus*). It showed the highest collagen yield between other proteases (papain, pepsin, and trypsin) with excellent chemical properties and antioxidant activity [38].

Furthermore, Elavarasan et al. [11] found that BR (10.62%) had the second-best DH (%) of water-washed catla meat in the optimum conditions after the commercial enzyme protamex (12.61%). In another case, Sadabpong et al. [39] found that the BR was between the best enzymes for the hydrolysis (50.59% DH) of gelatine extracted from Nile tilapia skin. Auwal et al. also investigated the hydrolysis of stonefish protein with BR, obtaining 54.62% of DH's optimum conditions [7]. In our case, although the DH did differ between the enzymes, with BR, proper digestion of the protein material is still obtained. In addition, the electrophoresis for both enzymes showed that BR is an excellent enzyme to obtain low molecular weight peptides from the collagen of cod skin. Low molecular weight collagen hydrolysates are generally thought to exert better bioactivities. However, many factors affect their preparation, such as the long-lived protein property, the strong and tough cross-links, and the presence of Hydroxyproline (Hyp) in the structure that plays a key role in collagen stability [13]. Possibly, since BR is a protease that acts by making random cuts, this property helps to increase the probability of obtaining low molecular weight hydrolysates with greater bioactivities.

Consequently, during the optimisation process, the high value of protein extraction and high antioxidant and antihypertensive activities in the cod collagen-rich enzymatic hydrolysate were found. The optimum conditions were pH 4.0 and temperature 28.9 °C during 30 min of hydrolysis. Previously, collagen peptides derived from fish waste were reported to have antioxidative activity [10,11,40,41] and antihypertensive activity [39,42–44]. The antioxidant activities of bioactive peptides are mainly due to some aromatic amino acids and histidine. Furthermore, fish collagen peptides are rich in hydrophobic amino acids because of the high percentage of Gly and Pro. These structural characteristics make the marine collagen peptides possess higher antioxidant effects than peptides derived from other proteins [45]. In the present work, we detected great Oxygen radical absorbance capacity (ORAC assay) and inhibitory activity on the angiotensin-I converting enzyme (iACE assay) in the collagen-rich enzymatic hydrolysate of SCCS treated with BR. In addition, if BR is not removed, our final product benefits from the anti-inflammatory properties already reported for this enzyme [46,47].

Regarding the biocompatibility of the SCCS hydrolysate, this showed to be not cytotoxic for any of the cell lines assayed. Other works suggested that fish collagen exhibits comparable biocompatibility to mammal's skin collagen, indicating it might be a potential alternative to type I collagen from mammals in many applications [48,49].

These results suggest that it was possible to obtain, with an enzymatic/mild acid process, collagen hydrolysate from SCCS with potential antioxidant and antihypertensive properties.

## 5. Conclusions

In the present study, we reported an integrated and sustainable methodology to obtain a collagen hydrolysate from SCCS using a plant enzyme as protease, BR. The enzymatic hydrolysis of SCCS suffered the influence of the variables process being the T (°C) and pH the most influent on the protein content, antioxidant, and antihypertensive activities. Optimal process conditions: pH of 4.0, the temperature of 28.9 °C, and time of 30 min, resulted in high antioxidant activity value (ORAC: 514 µmol Trolox Equivalent/g protein) and high iACE value (IC$_{50}$: 166 µg protein/mL). Furthermore, the biocompatibility assay showed that it was not cytotoxic for L929 and HaCaT cell lines. So, this is a sustainable process that consumes low time and energy and uses an alternative source as raw material. Thus, potential applications as a bioactive peptide source can be evaluated.

**Supplementary Materials:** The following are available online at https://www.mdpi.com/article/10.3390/app11188538/s1.

**Author Contributions:** Conceptualisation, E.R.C. and M.P.; methodology, E.R.C.; software, E.R.C. and M.E.B.; validation, E.R.C. and M.E.B.; investigation, E.R.C. and M.E.B.; resources, M.P.; writing—original draft preparation, E.R.C. and M.E.B.; writing—review and editing, E.R.C., M.E.B., and M.P.; supervision, E.R.C. and M.P.; project administration, M.P.; funding acquisition, M.P. All authors have read and agreed to the published version of the manuscript.

**Funding:** This research was funded by Foundation for Science and Technology (FCT) through the "MultiBiorefinery" Project (POCI-01-0145-FEDER-016403).

**Institutional Review Board Statement:** Not applicable.

**Informed Consent Statement:** Not applicable.

**Data Availability Statement:** The data presented in this study are available in Supplementary Materials.

**Acknowledgments:** The authors thank the company Pascoal & Filhos S.A. for kindly providing the salt-cured cod skin used in this study. Furthermore, the authors thank the CBQF for the institutional support.

**Conflicts of Interest:** The authors declare no conflict of interest.

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
