# Peer review of "Collagen-Based Bioactive Bromelain Hydrolysate from Salt-Cured Cod Skin"

_applsci, doi:10.3390/app11188538_

Round 1

Reviewer 1 Report

The paper is well written and the topic is interesting and has scientific and commercial value. The methods that the authors have used to test their hypothesis in both the text and the appendix are suitable and valid

The authors should refer to the recently published article and discuss it in the discussion (attached): Devita, Liza, Mala Nurilmala, Hanifah Nuryani Lioe, and Maggy T. Suhartono. "Chemical and Antioxidant Characteristics of Skin-Derived Collagen Obtained by Acid-Enzymatic Hydrolysis of Bigeye Tuna (Thunnus obesus)." Marine Drugs 19, no. 4 (2021): 222.

Overall, I recommend the manuscript for publication.

Author Response

We thank the Reviewer for the relevant comment. As the Reviewer suggested, we added the following sentences and reference to the section discussion in line 284:

“BR was also successfully applied to collagen extraction from the skin of bigeye tuna (Thunnus obesus). It showed the highest collagen yield between other proteases (papain, pepsin, and trypsin) with excellent chemical properties and antioxidant activity [37].”

Reference:

37.        Devita, L.; Nurilmala, M.; Lioe, H.N.; Suhartono, M.T. Chemical and Antioxidant Characteristics of Skin-Derived Collagen Obtained by Acid-Enzymatic Hydrolysis of Bigeye Tuna (Thunnus obesus). Mar. Drugs 2021, 19, 222.

Reviewer 2 Report

The present article aimed at optimising the production of bioactive peptides obtained from collagen-rich salt-cured cod skin using the protease Bromelain. The use of collagen-derived peptides from cod skin proved to be a promising alternative to the traditional bovine source. Interestingly, the peptidic fractions obtained from Bro activity over fish collagen were capable of inhibiting human angiotensin converting enzyme I, in a possible antihypertensive effect, as well as acting in an antioxidant manner. Finally, the authors showed the biocompatibility of the generated material using cell culture and defined the optimized conditions to obtain the best enzyme activity and biological effect.

Discussion. It is well-know that Bromelain is composed of different enzymatic isoforms. Do you think that the optimized pH enriches any specific isoform and that this difference reflects the observed biological effects?

Moderate grammar corrections are needed. A few examples:

line 49: instead of "in the past two decades led to academia and the industry explore", change to "in the past two decades led academia and the industry to explore".

line 42, 43: "collagen extraction takes is time and energy". Maybe rephrase this.

line 58: ...industry abundant "in in" countries such as Portugal...

Author Response

We are very grateful to the Reviewer for the comments, which helped enrich our work. The requested modifications can be seen with change control in the new version of the manuscript. Below are responses to the different comments from the Reviewer.

Discussion. It is well-know that Bromelain is composed of different enzymatic isoforms. Do you think that the optimized pH enriches any specific isoform and that this difference reflects the observed biological effects?

As we commented in a previous article “The pineapple plant contains at least five distinct cysteine proteases belonging to the papain family. The major protease present in pine- apple stem (heart and cylinder of the pineapple) is stem BR (EC 3.4.22.32) and the other minor protease include Ananain (EC 3.4.22.31), Comosain and SBA (acidic stem BR) (Maurer, 2001). Fruit BR (EC 3.4.22.33) is the major protease in the pulp.” In line 71 we had already specified that the isoform used was the stem bromalain. Below is the reference.

Reference:

Campos, D. A., Coscueta, E. R., Valetti, N. W., Pastrana-Castro, L. M., Teixeira, J. A., Picó, G. A., & Pintado, M. M. (2019). Optimization of bromelain isolation from pineapple byproducts by polysaccharide complex formation. Food Hydrocolloids, 87, 792–804. https://doi.org/10.1016/j.foodhyd.2018.09.009

Moderate grammar corrections are needed.

Thank you very much for the suggestion, we made a complete review and corrected the grammatical errors mentioned as well as others found.